# Towards a Near-Real-Time Protocol Tunneling Detector Based on Machine Learning Techniques †

Filippo Sobrero *, Beatrice Clavarezza , Daniele Ucci * and Federica Bisio

aizoOn Technology Consulting, 10146 Turin, Italy; beatrice.clavarezza@aizoongroup.com (B.C.);
federica.bisio@aizoongroup.com (F.B.)
* Correspondence: filippo.sobrero@aizoongroup.com (F.S.); daniele.ucci@aizoongroup.com (D.U.)
† This paper is an extension of our paper published in IEEE Symposium Series on Computational Intelligence,
Orlando, FL, USA, 5–7 December 2021.

**Abstract:** In the very recent years, cybersecurity attacks have increased at an unprecedented pace, becoming ever more sophisticated and costly. Their impact has involved both private/public companies and critical infrastructures. At the same time, due to the COVID-19 pandemic, the security perimeters of many organizations expanded, causing an increase in the attack surface exploitable by threat actors through malware and phishing attacks. Given these factors, it is of primary importance to monitor the security perimeter and the events occurring in the monitored network, according to a tested security strategy of detection and response. In this paper, we present a protocol tunneling detector prototype which inspects, in near real-time, a company's network traffic using machine learning techniques. Indeed, tunneling attacks allow malicious actors to maximize the time in which their activity remains undetected. The detector monitors unencrypted network flows and extracts features to detect possible occurring attacks and anomalies by combining machine learning and deep learning. The proposed module can be embedded in any network security monitoring platform able to provide network flow information along with its metadata. The detection capabilities of the implemented prototype have been tested both on benign and malicious datasets. Results show an overall accuracy of 97.1% and an F1-score equal to 95.6%.

**Keywords:** passive network analysis; DNS tunneling; anomaly detection; machine learning; deep learning

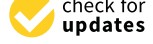



## 1. Introduction

Cybersecurity attacks keep increasing year over year at an unprecedented pace, becoming ever more sophisticated and costly [1,2]. The growth between 2021 and 2022 has resulted in a rise of attacks' volume and impact on both private/public companies and critical infrastructures. Companies comprise digital service providers, public administrations, and governments and include businesses operating in the finance and health sectors. In particular, service providers have experienced a raise of more than 15% in intrusions (infamous is the case of Solarwinds [3]) compared to 2021 [1], a trend destined to grow in the next years [4]. At the same time, due to the COVID-19 pandemic, the security perimeters of many organizations expanded to cope with the new needs of remote working, causing an increase in the attack surface exploitable by attackers [4]. The European Union Agency for Cybersecurity estimates that more than 10 terabytes of data are stolen monthly from target assets that are made unavailable, until a ransom is payed [1], while IBM calculates that the average cost of these attacks is USD 4.54 M, increasing up to USD 5.12 M [2]. On the other hand, malware attacks are still on the rise after the pause recorded during the pandemic, and phishing continues to be the common attack vector for initial access [1].

Given these factors, it is of primary importance to monitor the security perimeter and the events occurring in the network, according to a tested security strategy of detection

and response. According to Gartner [4], newly proposed solutions should be automated as much as possible, since human errors continue to play a crucial role in most security breaches. In this context, machine learning turned out to be a natural choice for automated analyses and prevention of this kind of threats [5]. The strength of machine learning lies in its ability to identify hidden patterns and correlations in large volumes of raw data and leverage such features to recognize previously unseen attacks. In this paper, we present a protocol tunneling detector prototype which inspects—in near real-time—a company's network traffic using machine learning. Tunneling techniques allow attackers to create a tunnel through a network by encapsulating traffic inside another protocol [6]; hence, it can be used to let infected machines contact their corresponding command-and-control centers. Thus, by abusing legitimate network traffic protocols, like DNS [7], the attacker maximizes the time in which the infection remains undetected. In this work, we rely on a commercial network security monitoring platform for detecting and investigating potentially malicious or anomalous activities [8–11], but the proposed solution can be easily integrated into any network security monitoring platform able to provide network flow information along with its metadata. The platform we employ is responsible for collecting, processing network flows, and dispatching them to one or more advanced cybersecurity analytics (ACAs) which are able to recognize the signals of possible occurring attacks and anomalies. In this scenario, the detector monitors only clear-text protocols, but it works jointly with an ACA responsible for analyzing encrypted traffic [11]. Indeed, while some clear-text protocols are extensively used (i.e., DNS), nowadays, the vast majority of Internet traffic is encrypted [12–16]: this enabled threat actors to perform malware campaigns relying on HTTPS for delivering malware and contacting command-and-control centers [17]. Just in 2020, 67% of malware has been delivered via encrypted HTTPS connections [18]. Along with malware delivery, malicious secure communications are used to exfiltrate data and steal sensitive information from private and public companies [19–21]. While the analytics dealing with encrypted traffic has been extensively described in [11], we extend this previous work by backing up secure connection analysis to the monitoring of clear-text protocols. As mentioned before, the latter can be used to discover the abuse of such protocols and signal network packets' contents which are not usually observed in the monitored network. The module presented in this paper extracts a sequence of $N$ bytes from each single network packet and computes features associated to the collected stream of bytes. Through the combination of deep learning and machine learning, each network packet is assigned to a specific network protocol; if a connection exhibits anomalies (e.g., an interleaving of different protocols), a security analyst is notified about the discovered inconsistency. More specifically:

- we implement a protocol tunneling detector prototype which analyzes, in near real-time, a byte sequence of the packets flowing in the monitored network.
- the proposed prototype combines
  - an artificial neural network (ANN), based on [22], that accurately classifies clear-text protocols and identifies possible anomalies in network connections;
  - a support vector machine that is able to detect compressed/encrypted traffic within unencrypted connections.
- we design and implement an input sanitization module, which automatically removes inconsistent data from models' training sets to significantly increase the models' performance.

With respect to [22], we changed both the input byte sequences we provide to the ANN and their sizes in bytes (as detailed in Sections 4.1 and 5). The performance of the proposed approach has been evaluated on different datasets that either contain legitimate traffic or simulate DNS tunneling attacks, which are the most common [7]. The obtained overall accuracy of the proposed prototype is 97.1%, along with an F1-score equal to 95.6%. It is worth noting that, being the prototype trained with only legitimate traffic, it is potentially able to identify zero-day attacks that deviate from the usual traffic observed in the network.

The rest of the paper is organized as follows: Section 2 discusses related work, while Section 3 introduces basic notions that will be later used to detail the proposed approach (Section 4). The experimental evaluation is reported in Section 5, followed by Section 6, where we discuss the strengths of our prototype and some key design choices we made. Finally, Section 7 concludes the paper.

## 2. Related Work

Tunneling attacks are a specific typology of network attacks in which an attacker creates a tunnel through a network by encapsulating traffic inside another protocol [6]. This allows the attacker to bypass traditional network security controls and potentially exfiltrate sensitive information. Therefore, as discussed in Section 1, using clear-text network protocols may pose a significant risk when these are abused by malicious actors. In this context, DNS tunneling represents one of the most common techniques employed for covertly exfiltrating data from a network, by encoding the data in DNS queries and responses. Since this method is becoming increasingly prevalent, a growing body of research aims at detecting and mitigating DNS tunneling attacks. In [23], the authors review detection technologies from a perspective of rule-based and model-based methods with descriptions and analyses of DNS-based tools and their corresponding features, covering detection approaches developed from 2006 to 2020 by means of a comparative analysis.

Latest works in the area of DNS tunneling detection mainly cover three main categories, i.e., detection approaches via machine learning, real-time detection approaches, and detection of DNS tunneling variants (e.g., fast flux [9] and domain generation algorithms (DGAs) [8]).

Regarding the first group, researchers have recently proposed both machine and deep learning algorithms for detecting DNS tunneling traffic, such as support vector machines (SVMs), random forests and Convolutional Neural Networks (CNNs) and Recurrent Neural Networks (RNNs), respectively. Do et al. have proposed an SVM to identify DNS tunneling attacks within mobile networks, by using features such as time, traffic source and destination, and length of DNS queries [24]. Other researchers have proposed a random forest classifier to detect this kind of attack [25]. They included in their features the number of answers provided by a DNS response and the time between two consecutive packets and responses for a specific domain. Random forests are also employed in hybrid solutions like the one proposed in [26], where a 100-trees random forest is paired with a CNN; they achieved good performance on their dataset, and it is worth noting that, during their experiments on traffic collected from a real network, they were able to identify a domain associated to a command-and-control center. In [27], the authors developed a novel DNS tunneling detection method employing a Convolutional Neural Network (CNN) to analyze DNS queries and responses and identify DNS tunneling activities. The proposed approach is evaluated using a dataset of real-world DNS traffic and shows promising results in detecting DNS tunneling attacks with high accuracy. The work of [28] applies both Convolutional Neural Networks (CNNs) and Recurrent Neural Networks (RNNs) for detecting DNS tunneling traffic. The authors have shown that these algorithms can effectively spot and identify malicious patterns.

The second group of studies has focused on developing real-time detection systems for DNS tunneling. These systems use a combination of several detection techniques to timely identify malicious DNS traffic [29]. In [30], the authors presented an overview of principal countermeasures for DNS tunneling attacks.

Regarding the state of the art of approaches that analyze encrypted communications, it has already been presented in [11].

The approach we present and evaluate in the next sections passively extracts both sequential and statistical features from network flows to detect tunneling attacks in clear-text protocols. As sequential features, we refer to those characteristics obtained from raw flow sequences. Most works rely on similar features, like domain-based features [23,29,30], including the domain name itself [27,28] and payload and volumetric features [23,30]. These can only be obtained when the entire packet has been reconstructed by a network

analyzer. Differently, for each packet, we directly examine a specific sequence of bytes without requiring to compute and store any packet-related metadata. In addition, we use artificial neural networks, which are simpler deep learning models and, hence, require less computing resources to be trained.

## 3. Background

### 3.1. DNS Tunneling

Protocol tunneling is an attack technique commonly used to maximize the time in which the infection remains undetected in a targeted network. In this context, the DNS protocol is usually abused in order to bypass security gateways and, then, to tunnel malware and other data through a client–server model [7]. Figure 1 depicts a typical DNS tunneling scenario: firstly, an attacker registers a malicious domain (e.g., attacker.com, accessed on 17 October 2023) on a C&C center managed by her; at that point, assuming that the attacker has already taken control over a machine inside the targeted network and violated its security perimeter, the infected computer sends a query to the malicious domain. Since DNS requests are typically allowed to move in and out of the network, the query through the DNS resolver reaches the attacker's C&C center, where the tunneling program is installed. This established tunnel can be used either to exfiltrate data and sensitive information or for other malicious purposes.

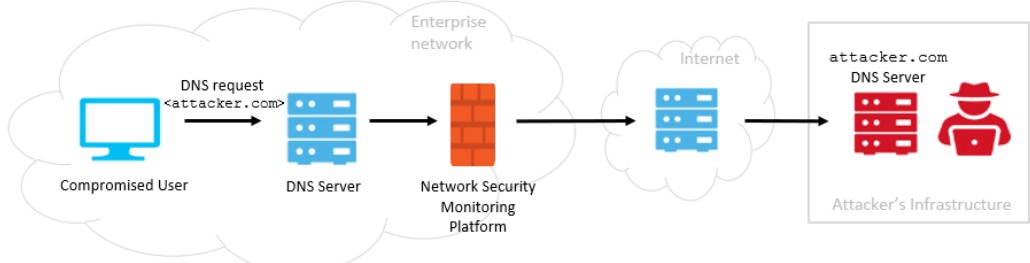

**Figure 1.** A DNS tunneling example.

### 3.2. Support Vector Machines

The original formulation of support vector machines [31] (SVMs) is related to the resolution of supervised tasks with the objective of finding a maximum margin hyperplane that separates two or more classes of observations. In the last years, one-class SVMs have also been shown to represent a suitable choice in the context of anomaly and outlier detection [32]. It is defined as a boundary-based anomaly detection method, which modifies the original SVM approach by extending it in order to deal with unlabeled data. Like traditional SVMs, one-class SVMs can also benefit from the so called kernel trick when extended to non-linearly transformed spaces, by defining an appropriate scalar product in the feature space.

### 3.3. Artificial Neural Networks

Artificial neural networks (ANNs) are deep learning models that have been successfully applied to a vast number of knowledge fields ranging from computing science to arts [33]. They are internally constituted by groups of multiple neurons, which can be thought of as mathematical functions that take one or more inputs. In ANNs, inputs are only processed forward and are multiplied by weights within each neuron and summed up to be then passed to an activation function, which becomes the neuron's output. In general, artificial neural networks consist of three different layers: input, hidden, and output; the first layer accepts inputs, while the hidden layers process them to learn the optimum weights. Finally, the output layer produces the result.

## 4. Protocol Tunneling Detector

The proposed architecture splits the burden of processing the traffic of a monitored network into two different sub-modules: the first mainly deals with secure connections,

while the second inspects unencrypted traffic. As previously discussed, the former analytics has been detailed in [11]. At a glance, it detects possible anomalies occurring during an SSL/TLS handshake between a client, located inside the network monitored by the software platform outlined in Section 1, and an external server. The SSL/TLS detection analytics examines information contained in X.509, SSL, and TLS exchanged protocol messages. Instead, the second module looks for anomalies in unencrypted traffic regarding the abuse of specific protocols (i.e., tunneling attack techniques). To provide these detection capabilities, this prototype collects a sequence of bytes from each network packet and inspects its content. The content, along with its features, is fed to a testing module, which detects possible anomalies that are signaled to security analysts.

## 4.1. General Approach

Figure 2 reports the general structure of the proposed anomaly detection methodology, which runs in near-real-time fashion. Indeed, a delay is introduced both by data processing and anomaly evaluations that are not performed on the single packet but, rather, on the entire connection, meaning that the approach has to wait to have enough information to make a decision. Hence, for each packet observed in the live network traffic, the prototype collects a sequence of $N$ bytes belonging to the highest network protocol used in the communication. As an example, in a secure connection which relies on HTTPS, the bytes returned by the extraction process are the ones related to HTTPS, and not to the other lower-layer protocols (e.g., TCP).

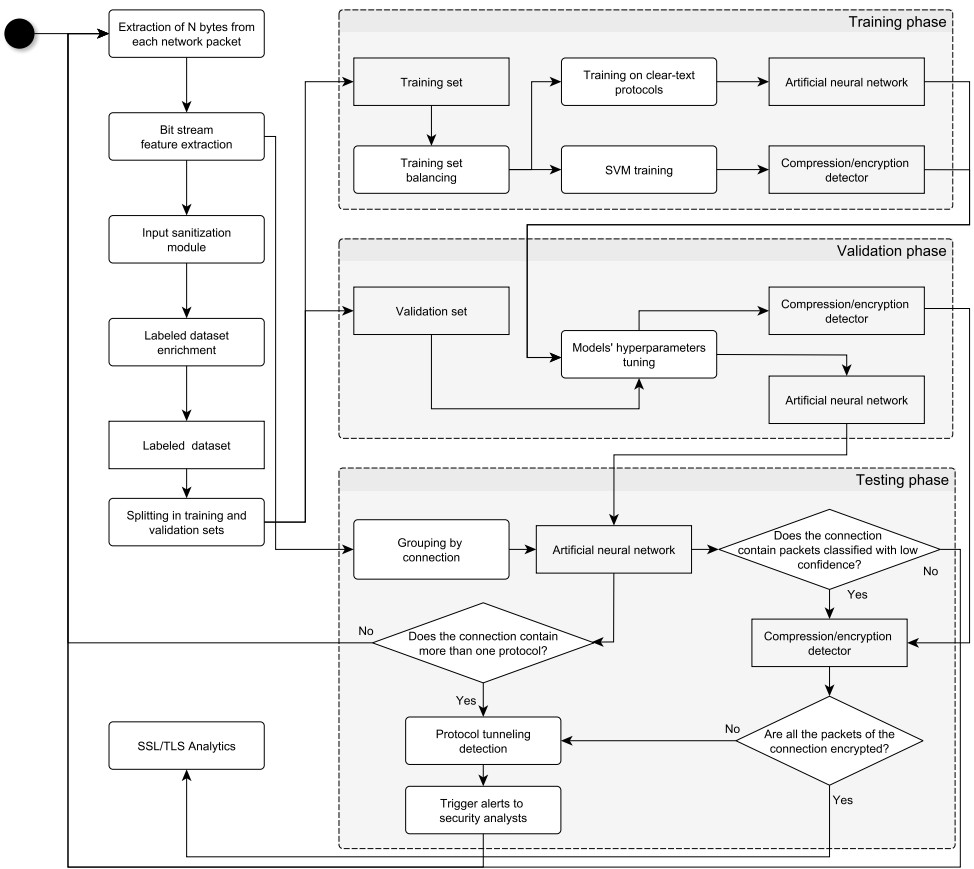

**Figure 2.** Protocol tunneling detector prototype overview.

From the obtained bit stream, we extract the following sequential features (i.e., those features obtained from raw flow sequences):

- binary representation of collected bytes

- bit-stream entropy and *p*-values obtained from statistical tests for random and pseudorandom number generators for cryptographic applications [34]
- statistical properties of the bit-stream hexadecimal representation

and we keep the protocol label associated to the bit stream itself. While the binary representation of the *N* bytes is meant to label the protocol of each packet under analysis, the sequential features allow to understand if the packet content is either compressed or encrypted.

After feature extraction, the raw dataset constituted by streams of bits and their corresponding labels is properly sanitized. Indeed, it is easily possible to lightly label the network packets belonging to a connection by simply looking either at the ports or at the connection metadata. However, this labeling may be prone to errors since it either does not take into account potential custom configurations of services (e.g., SMB protocol operating on a port different from 445) or intentional misuse of specific protocols by attackers (as in the case of tunneling). Moreover, clear-text protocols may transfer packets containing compressed data, whose presence could compromise the identification of the correct network protocol. Hence, it is paramount to have a refined and clean dataset to let models perform at their best. During our experimental evaluations, we have found out that the accuracy of the trained models, after refining the raw dataset, has significantly increased: 7% for the ANN and 20% for the compression/encryption detector.

To achieve this performance boost, we have specially implemented an input sanitization module, shown in Figure 3. In this module, we combine unsupervised and supervised support vector machines (SVMs) to clean the raw dataset: first, for each network protocol, we train a one-class SVM both on clear-text and encrypted protocols in order to filter out outliers from the raw dataset. As an example, in protocols like HTTP and SMB, requests and responses may contain either the content of (compressed) files or other types of information that are not strictly correlated with the specific protocol communication patterns. Thus, in order to exclude these outliers, we build one-class SVMs, one for each different protocol, whose hyperparameters are properly tuned on the raw labeled dataset. Trained models are then applied to identify outliers and remove them from the raw dataset. This refined dataset is then used to train an SVM by applying a one-vs-all classification for detecting packets which are either compressed or encrypted. This single classifier is applied to remove both compressed and encrypted packets from clear-text protocols. It is worth mentioning that, in proxied environments, encrypted packets may be present in connections labeled as HTTP: indeed, in these scenarios, secure communications also pass through the proxy, even if these connections are erroneously labeled as HTTP. As already outlined in Section 3, one-class SVMs are successful in identifying outliers; for this reason, we have extensively used them to sanitize our training sets with remarkable results.

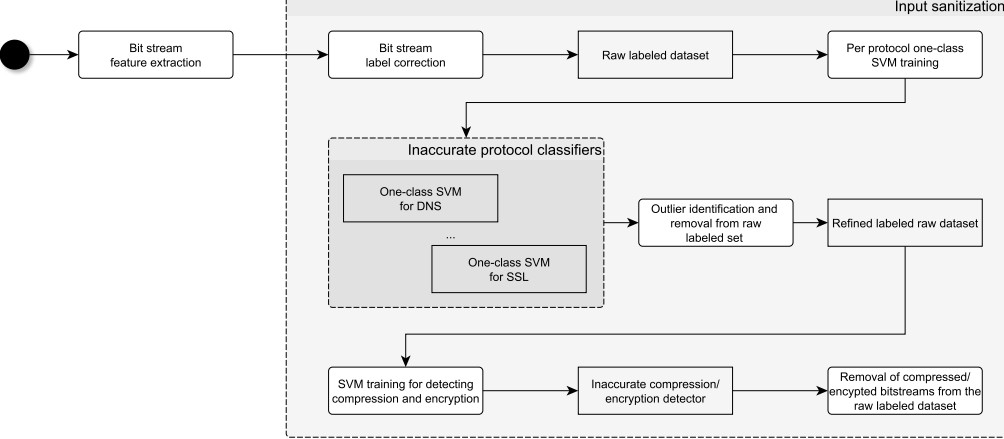

**Figure 3.** Input sanitization module.

This sanitized dataset is then split into training and validation sets to essentially build two different models: (i) an artificial neural network (ANN) able to classify clear-text protocols (e.g., DNS) and (ii) an SVM that is a compression/encryption detector for identifying, respectively, compressed and encrypted packets. As later shown in Section 5, after construction, the training set is considerably unbalanced towards secure protocols. For this reason, we apply the SMOTE data augmentation technique [35] to increase the samples of those protocols belonging to minority classes. During the test phase, performed light labeling based on connection's destination port is not taken into account, and the resulting bit streams are grouped by connection. Each packet is given in input to a trained ANN (whose training process is detailed in Section 4.3) and the analytics both verifies if, in the connection, there are some packets that have been classified with low confidence and more than one protocol is present. While in this latter case, the co-presence of multiple protocols might signal a possible tunneling attack, when the ANN classifies packets with low confidence, then, the connection could contain either compressed/encrypted packets or packets whose byte sequences differ from the ones usually observed in the network. To distinguish between these two cases, a more in depth verification is carried out: if the connection is not entirely encrypted, meaning that it is a not a secure communication, the prototype checks if the packets signaled as anomalous (i.e., with low confidence) by the ANN are either encrypted or belongs to another protocol. If either encryption or compression is detected, the anomaly is notified to security analysts. On the other hand, if the entire connection is encrypted, it is collected and stored in a database, periodically accessed in order to retrieve data and metadata about X.509, SSL, and TLS exchanged protocol messages in order to be analyzed by the analytics described in [11]. As outlined earlier, all the compression/encryption tests are performed using an SVM, capable of correctly classifying network packets, but the proposed classifier could be substituted with other valid alternatives, such as random forest models.

### 4.2. Feature Extraction

As discussed in Section 4.1, sequential features allow us to understand if the content of a network packet is either compressed or encrypted. We rely on a statistical package developed by the Information Technology Laboratory at the National Institute of Standards and Technology, containing a set of 15 tests that measure the randomness of a binary sequence [34]. These tests have been designed to provide a first step towards the decision whether or not a generated binary sequence can be used in cryptographic applications, namely if the sequence appears to be randomly generated. In other words, each new bit of the sequence should be unpredictable. From a statistical point of view, each test verifies if the sequence being under analysis is random. This null hypothesis can be either rejected or accepted depending on the statistic value on the data exceeding or not a specific value—called critical value—that is typically far in the tails of a distribution of reference. Test reference distributions used in the NIST tests are the standard normal and the $\chi^2$ distributions. Even if the statistical package contains 15 tests, we use only 5 of them, because the length $N$ of the binary sequence we test does not meet the corresponding input size recommendation in [34]. To each sequence, we apply the following tests: frequency within a block, longest-run-of-ones in a block, serial test, approximate entropy, and cumulative sums. In addition, in our experimental evaluations, we extract some statistical properties and compute the Shannon entropy metrics [36] that, combined with the previously mentioned tests, have shown to improve the overall accuracy of the classification. As statistical properties, the following features are extracted from the corresponding hexadecimal representation $h$ of a bit stream of $N$ bytes:

- number of different alphanumeric characters in $h$ normalized over $h$ length;
- number of different letters in $h$ normalized over $h$ length;
- longest consecutive sequence of the same character in $h$ normalized over $h$ length.

### 4.3. Input Sanitization

For accurately training machine learning models, the training set should be as much "clean" as possible. In Section 4.1 we have already discussed how labeling based on connection metadata could be error prone either due to potential custom configurations of services, intentional misuse of specific protocols by attackers, or network protocols encapsulating compressed data. In addition, during our experimental evaluations, we have observed that in some cases the employed traffic analyzer can assign an empty label or multiple labels to a single network packet. While in the first case bit streams with empty labels can be easily discarded for the training phase, in the presence of multi-labels, it is possible to assign a unique correct label if a protocol that is monitored by the prototype itself exists among the labels. As an example, if the assigned labels are NTLM, GSSAPI, SMB, and DCE_RPC, the resulting label is SMB. For these reasons the very first step of the sanitization module is to correct the multi-labels associated to bit streams and discard the empty ones. Then, we train an ensamble of one-class SVMs, one for each protocol (see Figure 3): each different classifier is properly tuned to filter out outliers from the raw dataset. As stated in Section 4.1, HTTP and SMB requests or responses may contain either the content of (compressed) files or other types of information that are not strictly correlated with the specific protocol communication patterns. Trained models are then applied to identify these kinds of network packets, and they are removed from the raw dataset. This preprocessed dataset is used to train a supervised support vector machine, called compression/encryption detector, by applying a one-vs-all classification for detecting packets which are either compressed or encrypted. It is worth noting that all these models are still inaccurate because they are trained on a "dirty" dataset. Hence, to further increase the quality of the labels and obtain the final training set, the compression/encryption detector is fed with clear-text bit streams to remove possible compressed/encrypted packets from clear-text protocols, as in the case of proxied environments. The result of this sanitization process is a dataset which allows to train and validate two accurate models: an artificial neural network for clear-text protocols and an SVM for compressed and encrypted traffic.

### 4.4. Anomaly Detection

During the test phase (see Figure 2), bit streams are analyzed by the trained ANN. In turn, the ANN flags three different cases as potential tunneling attacks and alerts security analysts when these cases occur: (i) the high confidence detection of more than one protocol in the same connection, (ii) the low confidence detection of one protocol for all the packets in the same connection, and (iii) the labeling, both with high and low confidence, of one or more protocols for the packets belonging to the same connection (as in the case of secure protocols over DNS). As later specified in Section 5, in the ANN, the high/low confidence threshold $c$ can be dynamically set. In any case, the detection of encrypted packets into a clear-text connection generates alert notifications enriched with the information about the presence of encrypted protocol messages. Possibly, notified alerts can be filtered whitelisting source and/or destination IPs to reduce the false positives caused by well-known machines.

Hence, if some packets of the connection are classified with low confidence, the corresponding bit stream's sequential features (refer to Section 4.2) are given in input to the compression/encryption detector. If all the packets contained in the connection are encrypted, then the connection and its corresponding metadata are given in input to the SSL/TLS analytics for further scrutiny [11]. On the contrary, if the connection contains some compressed/encrypted packets or none of them, depending on the protocol, the connection is considered anomalous. Indeed, it is worth noting that the combination of two different protocols is not always a signal of an occurring attack: as already discussed, SMB and HTTP connections can contain protocol-specific messages along with compressed data; however, DNS messages interleaved with other protocols are highly suspicious. Finally, since each single module of the proposed prototype has been trained only with legitimate traffic, it is potentially able to spot zero-day attacks having features which are different from the ones usually observed in the network.

## 5. Experimental Evaluation

The proposed prototype and the experimental evaluations have been, respectively, implemented and performed in Python. The size $N$ we have chosen for the byte sequences, extracted from network packets, is 52 bytes. More in detail, we retrieve the first 64 bytes of the payload of each TCP/UDP packet, from which we remove the first 12 B: indeed, a preliminary evaluation has shown that these first bytes had a very low variance in their binary representation among different packets of the same protocol. The specific selection of the byte sequence to extract has improved the accuracy of the trained neural network, increasing its anomaly detection capabilities.

For the experimental evaluation of the proposed prototype, we collected both benign and malicious datasets. The benign communication dataset contains a subset of legitimate traffic observed in a real corporate network during a period of about 2 days. From this initial dataset, we sample connections to start building the models training sets and the dataset that will be used for testing. Figure 4 summarizes general statistics about the collected training set in terms of packets, before and after sanitization, while Table 1 reports how the test set of legitimate network traffic is characterized. The sanitization process makes the training set, which is obviously unbalanced towards encrypted protocols, balanced: indeed, after sanitization, the number of packets belonging to, respectively, clear-text and secure protocols is almost even. It is worth noting that the balanced training set for the ANN, containing DHCP, DNS, NTP, HTTP, and SMB packets, also comprises data belonging to the KRB network protocol (i.e., encrypted): our experimental evaluations have shown that during the test phase, the neural network performs better when it is also trained with encrypted byte sequences. As an ANN, we use a Keras sequential model with three hidden layers. The input layer accepts 416 bits (i.e., 52 B) and the output layer consists of six neurons, one for each clear-text protocol and KRB. Regarding SVMs, we rely on the open-source library scikit-learn. For completeness, we report in Table 2 the hyperparameters we have used to train the different SVMs in the sanitization module; in addition, we also report the hyperparameters we obtained by tuning the compression/encryption detector in the validation phase. It is worth mentioning that the parameter $t$, in Table 2, is used for each protocol one-class SVM as a threshold to filter only those outliers which have a Shannon entropy greater than $t$.

The intuition behind this filtering is that byte sequences having high entropy do not specifically belong to clear-text protocol communications; thus, they have to be discarded from the training set.

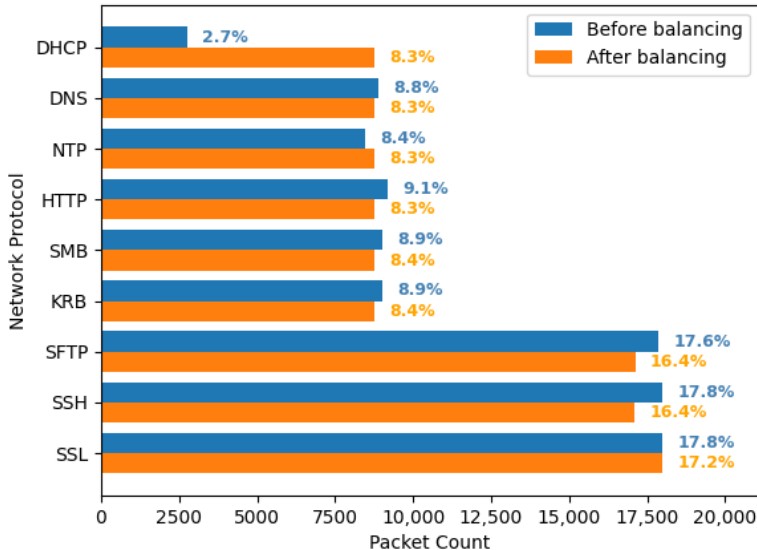

**Figure 4.** Packet distribution for each network protocol, before and after balancing.

**Table 1.** Benign test set composition.

| Statistics | Count [(%)] |
| --- | --- |
| DNS packets | 30,669 (1.10%) |
| SMB packets | 65,944 (2.35%) |
| HTTP packets | 262 (0.01%) |
| NTP packets | 46 (0.002%) |
| DHCP packets | 20 (0.001%) |
| KRB packets | 741 (0.03%) |
| SFTP packets | 69,158 (2.46%) |
| Not labeled packets | 61,552 (2.20%) |
| SSL packets | 2,571,608 (91.84%) |
| Distinct connections | 51,459 |
| Distinct source machines | 758 |
| Distinct dest. machines | 1566 |

On the other hand, malicious datasets are constituted by packet captures (PCAPs) shared by [37–39]. The former dataset contains three different types of DNS tunnels generated in a controlled environment, whose sizes are approximately 750 MB each. Tunneled data contain, respectively, SFTP, SSH, and Telnet malicious protocol messages. Each sample is made up of one single connection containing millions of DNS packets. It is reasonable to note that such connections would either easily stand out to security analysts or be simply detectable through well-known statistical approaches (e.g., outlier detection). Subsequently, as stated in Section 4.1, our approach groups data by connection; therefore, a single malicious packet is enough to flag the entire connection as anomalous. For the above reasons, we have decided to split each sample in $n$ different connections, composed by approximately 5000 DNS packets each. The size of the split, reported in Table 3, has been chosen according to the size of the connections monitored in the controlled environment. The second malicious dataset, instead, was born by the collaboration between the Bell Canada company's Cyber Threat Intelligence group and the Canadian Institute for Cybersecurity.

**Table 2.** Support vector machine hyperparameter settings.

| Model | Kernel | $\gamma$ | $\nu$ | $t$ | $C$ |
| --- | --- | --- | --- | --- | --- |
| DHCP one-class SVM | RBF | 0.7 | 0.03 | 0.77 | – |
| DNS one-class SVM | RBF | 0.7 | 0.03 | 0.77 | – |
| NTP one-class SVM | RBF | 0.03 | 0.1 | 0.92 | – |
| HTTP one-class SVM | RBF | 0.08 | 0.07 | 0.91 | – |
| SMB one-class SVM | RBF | 0.06 | 0.08 | 0.77 | – |
| KRB one-class SVM | RBF | 0.04 | 0.05 | 0.97 | – |
| SFTP one-class SVM | RBF | 0.7 | 0.05 | 0.97 | – |
| SSH one-class SVM | RBF | 0.7 | 0.05 | 0.97 | – |
| SSL one-class SVM | RBF | 0.0001 | 0.0028 | 0.97 | – |
| Compression/encryption detector | RBF | 0.01 | – | – | 100 |

In this dataset, we only take into account DNS packets that, in their payloads, contain exfiltrations of various types of files and we discard legitimate traffic. Moreover, it is worth mentioning that all the packets contained in [38] have been truncated at capture time to 96 B; this has required a slightly different approach to test these samples that will be discussed

later in this Section. Finally, [39] is a single packet capture to test detection and alerting capabilities of Packetbeat, Elastic's network packet analyzer. Malicious packet captures have been injected into the network security platform in order to be processed and analyzed as ordinary traffic. Table 3 reports a summary of the malicious assembled datasets: for each PCAP, we list the number of packets in the capture and which of these packets have been successfully processed by the platform's network analyzer (i.e., those packets whose size is greater or equal than 64 B); in addition, Table 3 depicts the number of connections in the PCAP and the number of them that have been identified as protocol tunneling attacks (i.e., true positives $TP$). Finally, the true positive rate $TPR$ of the proposed detector is reported for each packet capture. Analogously, Table 4 reports the same information contained in Table 3, but with reference to the test set described in Table 1. Being legitimate traffic, the last two columns report the connections mistakenly classified as tunnels (i.e., false positives $FP$) and the false positive rate $FPR$. The results of the evaluation, reported in Tables 3 and 4, show a false positive rate and a true positive rate, respectively, equal to 5.8% and 96.6%. The overall accuracy of the proposed prototype is 97.1%, while the resulting F1-score is 95.6%.

We conclude this section by discussing how we slightly modified the proposed approach, used in the other datasets, to be compliant with [38]. Indeed, the DNS packets contained in this dataset have been truncated during traffic acquisition, resulting in byte sequences that do not have the same length. In order to solve this dataset generation problem, we reduced all the DNS packets to a common length of 44 B, discarding the shorter byte sequences and trimming the longer ones. The result of the filtering operation is clearly shown in Table 3, where the number of processed PCAP packets is more than 54% less than the ones received in input by the traffic analyzer.

Since the bit-stream lengths are different from the datasets [37,39], we retrained our ANN to be fed with 44 B sequences. On the contrary, for this evaluation, we maintained the same hyperparameters for the different SVMs, reported in Table 2, and the same threshold $c$, used in the other experiments. In particular, for all our experimental evaluations, we set $c$ to 0.999999 in order to maximize the algorithm sensitivity and to compensate for the lesser information provided by the processing of [38]. This explains why, in the experimental evaluations, we were not able to achieve a very low false positive rate, as shown in Table 4.

**Table 3.** Malicious test set summary.

| Tunnel Type | No. of PCAP Packets | No. of Processed PCAP Packets | No. of Connections | $TP$ | $TPR$(%) |
|---|---|---|---|---|---|
| Telnet over DNS tunnel [37] | 2.4 M | 2.2 M | 457 | 457 | 100% |
| SFTP over DNS tunnel [37] | 2 M | 1 M | 209 | 209 | 100% |
| SSH over DNS tunnel [37] | 2.8 M | 2.7 M | 545 | 545 | 100% |
| Light file exfiltration [38] | 187,500 | 102,000 | 7617 | 7361 | 96.6% |
| Heavy file exfiltration [38] | 1.34 M | 765,000 | 43,964 | 42,441 | 96.5% |
| Data exfiltration over Iodine DNS tunnel [39] | 438 | 247 | 1 | 1 | 100% |

**Table 4.** Benign test set summary.

| Dataset | No. of PCAP Packets | No. of Processed PCAP Packets | No. of Connections | $FP$ | $FPR$(%) |
|---|---|---|---|---|---|
| Legitimate traffic | 5.4 M | 2.8 M | 51,459 | 2966 | 5.8% |

However, in context where a high number of false positives could be detrimental, $c$ can be tuned to obtain a 0.5% false positive rate or lesser without losing accuracy on protocol tunneling attacks.

## 6. Discussion

One of the most relevant challenges in cybersecurity is the detection of zero-day attacks, which can easily evade all the products based on signature or pattern detection. The proposed approach leverages various characteristics that are known to perform well when facing zero-day threats [40] like, for example, the absence of malicious samples in the training set, the training set sanitization process, and the absence of signature-based features and filters.

On the other hand, in Section 4.4, we suggested the usage of whitelists as a way of reducing false positives. While in the experimental evaluation of Section 5, we intentionally used them as little as possible (i.e., only 11 of the 758 machines in the benign test set were actually whitelisted), a security analyst could customize such whitelists in order to filter out machines that do not require monitoring. Adding domain knowledge to machine learning algorithms in the form of data (in our context, machines) that should not be modeled or monitored can not only reduce the amount of alerts that an analyst has to evaluate, but also increase model performance. In conjunction with the integration of whitelists, the number of false positives generated by our approach can be tuned in two other ways. The first one is represented by the threshold $c$ which, as already described in Section 5, controls the sensitivity of the ANN; in turn, it impacts the false positive rate because the lower the minimum value of the ANN output confidence considered as "high", the harder to match the conditions we have defined for the connection to be an anomaly (see Section 4.4). The second way of reducing false positives is a periodic retraining of proposed models. As briefly described in Section 7, once the prototype will be included in a streaming architecture, the training phase will be performed periodically. Real networks changes over time, so keeping the models updated is the key to maintain an accurate modeling of what is the current state of the network.

Finally, as already pointed out in Section 3, differently from other approaches, we extract features directly from the raw traffic without relying on network analyzers that reconstruct network traffic metadata. This allows to save computing resources and to speed up the analyses. Furthermore, the use of bit-stream representation is independent from protocol specific fields (e.g., DNS query field), making the prototype also able to detect tunneling attacks on different clear-text protocols.

## 7. Conclusions

In this paper, we proposed a software prototype for detecting protocol tunneling attacks in a monitored network. Relying on a combination of machine learning and deep learning techniques, the proposed solution identifies anomalous connections that deviate from the ones usually established in the network. Since machine learning models are only built based on legitimate traffic, the proposed solution is therefore able to deal with zero-day attacks, because malicious traffic is not required for the learning phase. The prototype has been evaluated both on malicious and benign datasets: results show a very high accuracy in detecting malicious samples and a low false positive rate on legitimate traffic.

As future work, we plan to optimize the algorithm through a deeper analysis on how the choice of byte-stream length affects the computational time, in order to find a value which guarantees the best trade-off between efficiency and accuracy. Indeed, in this work, we mainly focused on accuracy. Secondly, we envision that the engineered prototype will be integrated into a streaming architecture, where new data will be analyzed by the proposed prototype as soon as they are collected to provide the fastest possible response. In parallel, the models of the protocol tunneling detector are periodically retrained to keep them up-to-date with possible deviations from the usual behaviour of the monitored network. It is important to mention that the envisioned streaming architecture can always count on a trained model to process incoming traffic during possible retrainings; old models will be available until the new ones are ready. Finally, in Section 6, we discussed the benefits of IP whitelisting filters. Once in production, the prototype can be easily extended with other SOC-defined whitelists (e.g., whitelists regarding domains and/or

autonomous systems), allowing security analysts to enrich the proposed detector with their domain-specific knowledge, further reducing possible false positives and improving the overall performance.

**Author Contributions:** Conceptualization, F.S., D.U. and F.B.; methodology, F.S. and D.U.; software, F.S. and B.C.; validation, F.S., B.C., D.U. and F.B.; formal analysis, F.S. and B.C.; investigation, F.S. and B.C.; resources, D.U. and F.B.; data curation, F.S. and B.C.; writing—original draft preparation, F.S., B.C, D.U. and F.B.; writing—review and editing, F.S., B.C. and D.U.; supervision, D.U. and F.B. All authors have read and agreed to the published version of the manuscript.

**Funding:** This research received no external funding.

**Data Availability Statement:** Datasets containing DNS tunneling attacks can be found here: https://s3.eu-central-1.wasabisys.com/dns-tunneling/dns_tunnel_sftp.pcapng, https://s3.eu-central-1.wasabisys.com/dns-tunneling/dns_tunnel_ssh.pcapng, https://s3.eu-central-1.wasabisys.com/dns-tunneling/dns_tunnel_telnet.pcapng, https://www.unb.ca/cic/datasets/dns-exf-2021.html, and https://github.com/elastic/examples/blob/master/Security%20Analytics/dns_tunnel_detection/dns-tunnel-iodine.pcap, all accessed on 17 October 2023. Regarding the dataset containing the legitimate communications observed in a real corporate network, it is owned by aizoOn Technology Consulting and cannot be made available due to company policies.

**Conflicts of Interest:** The authors declare no conflict of interest, given that aizoOn Technology Consulting has not interfered with their ability to analyze and interpret data. Moreover, for this research, authors have not received any additional grant or funding.

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
