# Peer review of "Towards a Near-Real-Time Protocol Tunneling Detector Based on Machine Learning Techniques"

_jcp, doi:10.3390/jcp3040035_

Round 1

Reviewer 1 Report

Comments and Suggestions for Authors

This paper explores the significance of network security monitoring in the identification and mitigation of cyberattacks. It presents a prototype of a protocol tunnelling detector that employs machine learning techniques for the analysis of network data in order to identify and detect tunnelling assaults. The architectural design under consideration comprises two distinct sub-modules that are dedicated to the analysis of secure connections and unencrypted communications, respectively. The efficacy of the prototype is assessed through the utilisation of both benign and malicious datasets, demonstrating a acceptable level of accuracy in the identification of malicious samples.

The paper presents a protocol tunnelling detector prototype that uses machine learning techniques to analyze network traffic in near real-time. Tunnelling attacks, such as DNS tunnelling, can be used to bypass security measures and allow malicious activity to go undetected. 

Following are some of the observations,

1. Discuss more about Zero-Day Attack Detection; how the proposed prototype can detect zero-day attacks without relying on malicious traffic during the learning phase.

2. How to maintain the Low False Positive Rate

3. Discuss about the procedure of whitelisting of the proposed prototype.

4. The authors may highlight the complexities and considerations in applying machine learning to cybersecurity to build robust and effective detection systems.

5. The paper lacks analytical discussions on the effectiveness of proposed prototype w.r.t. the empirical results. A seperate section on this before the conclusion can be added.

Comments on the Quality of English Language

This paper explores the significance of network security monitoring in the identification and mitigation of cyberattacks. It presents a prototype of a protocol tunnelling detector that employs machine learning techniques for the analysis of network data in order to identify and detect tunnelling assaults. The architectural design under consideration comprises two distinct sub-modules that are dedicated to the analysis of secure connections and unencrypted communications, respectively. The efficacy of the prototype is assessed through the utilisation of both benign and malicious datasets, demonstrating a acceptable level of accuracy in the identification of malicious samples.

The paper presents a protocol tunnelling detector prototype that uses machine learning techniques to analyze network traffic in near real-time. Tunnelling attacks, such as DNS tunnelling, can be used to bypass security measures and allow malicious activity to go undetected. 

Following are some of the observations,

1. Discuss more about Zero-Day Attack Detection; how the proposed prototype can detect zero-day attacks without relying on malicious traffic during the learning phase.

2. How to maintain the Low False Positive Rate

3. Discuss about the procedure of whitelisting of the proposed prototype.

4. The authors may highlight the complexities and considerations in applying machine learning to cybersecurity to build robust and effective detection systems.

5. The paper lacks analytical discussions on the effectiveness of proposed prototype w.r.t. the empirical results. A seperate section on this before the conclusion can be added.

Reviewer 2 Report

Comments and Suggestions for Authors

In this paper, authors present a protocol tunneling detector prototype which inspects, in near real time, a company’s network traffic using machine learning techniques. The authors have also validated the superiority of using ML models in near real time prototype to detect anomalies.  In general, the paper's technical contributions are suitable for publications. The organization structure is also acceptable. Some enhancements should be made in terms of presentation especially the figures. Therefore I recommend minor revision.

Some  comments:

- I am not sure about near real time features of the prototype. Can you add more drawings how data is analyzed in near real time in your implementation of the prototype in Section 5?

- Figure 2 is not visible and should be enhanced further.

- Similar comments for figure 3 as well.

- On page 7 line 300 it should be "be dynamically set."

- What are the motivations of using basic ML models such as SVM? Why did not you use more advanced techniques such as transformers, etc. instead of ANN? More discussions on this aspect are needed.

- Can you show fig 4 y- axis in percentages as well so that it can be compared with Table 1?

Comments on the Quality of English Language

The English language is ok.

Reviewer 3 Report

Comments and Suggestions for Authors

In this manuscript, the authors propose present a protocol tunneling detector prototype which inspects, in near real time, a company’s network traffic using machine learning techniques. The detector monitors unencrypted network flows and extracts features to detect possible occurring attacks and anomalies, by combining machine learning and deep learning. The proposed module can be embedded in any network security monitoring platform able to provide network flow information along with its metadata. This work is interesting and has both theoretical and practical significance. However, there still exist some problems need to be necessarily addressed.

1、In the section of Related Work, some more previous work should be described.

2、Some figures are not clear and they should be improved, such as Figure 2, Figure 3, and so on.

3、The authors should provide some theoretical analysis to prove the superiorities of their proposed scheme.

4、It is better to describe the future work as a single section.

5、There are some grammatical mistakes and clerical errors.

Comments on the Quality of English Language

There are some grammatical mistakes and clerical errors. Then, the authors should improved them.

Reviewer 4 Report

Comments and Suggestions for Authors

The paper proposes an innovative and interesting approach to detect protocol tunnels in near-real-time mainly based on SVM and ANN techniques.

The proposed solution is very clear and well-described in detail in each single block. The obtained results well-assess the proposed solution showing good performance achievements.

However, the only major point that I raise is related to the complete lack of comparison with other solutions from the literature. Proposing something new that works well is important, but it is also important to show how it performs compared with other solutions already available in the literature. This point should be addressed including:

1.     A paragraph in Section 2 that highlights the similarities and differences between the proposed solution and the others mentioned in the section.

2.     Additional results and a related discussion in Section 5 to show how the proposed solution performs compared to some of the others from the literature in terms of the same performance metrics.

I also suggest, as a final minor remark, to increase the size of Figures 2 and 3. Even if the block logic is well explained within the text, it is almost impossible to read inside each block.

Round 2

Reviewer 4 Report

Comments and Suggestions for Authors

The authors addressed my raised points reporting satisfactory explanations about the made modifications and choices. The paper is ready to be published.